# A Systematic Review of Scientific Studies and Case Reports on Music and Obsessive-Compulsive Disorder

**DOI:** 10.3390/ijerph182211799

**Published:** 2021-11-10

**Authors:** Thanh Phuong Anh Truong, Briana Applewhite, Annie Heiderscheit, Hubertus Himmerich

**Affiliations:** 1Department of Psychological Medicine, Institute of Psychiatry, Psychology, and Neuroscience, King’s College London, London SE5 8AF, UK; anh.t.truong@kcl.ac.uk (T.P.A.T.); briana.applewhite@kcl.ac.uk (B.A.); 2Mental Health Studies Program, Institute of Psychiatry, Psychology, and Neuroscience, King’s College London, London SE5 8AB, UK; 3Department of Music Therapy, Augsburg University, Minneapolis, MN 55454, USA; heidersc@augsburg.edu; 4South London and Maudsley NHS Foundation Trust (SLaM), London SE5 8AZ, UK

**Keywords:** obsessive compulsive disorder, OCD, music, music therapy, musical obsession, musical hallucination, pseudohallucinations, involuntary musical imagery

## Abstract

Obsessive-compulsive disorder (OCD) is a severe psychiatric disorder, which can be associated with music-related symptoms. Music may also be used as an adjunct treatment for OCD. Following the PRISMA guidelines, we performed a systematic literature review exploring the relationship between music and OCD by using three online databases: PubMed, the Web of Science, and PsycINFO. The search terms were “obsessive compulsive disorder”, “OCD”, “music”, and “music therapy”. A total of 27 articles were utilised (*n* = 650 patients/study participants) and grouped into three categories. The first category comprised case reports of patients with musical obsessions in patients with OCD. Most patients were treated with selective serotonin reuptake inhibitors (SSRIs) or a combination of an SSRI and another pharmacological or a non-pharmacological treatment, with variable success. Studies on the music perception of people with OCD or obsessive-compulsive personality traits represented the second category. People with OCD or obsessive-compulsive personality traits seem to be more sensitive to tense music and were found to have an increased desire for harmony in music. Three small studies on music therapy in people with OCD constituted the third category. These studies suggest that patients with OCD might benefit from music therapy, which includes listening to music.

## 1. Introduction

### 1.1. Obsessive Compulsive Disorder: Symptoms and Diagnosis

The characteristics of obsessive-compulsive disorder (OCD) include repetitions of thoughts, images or impulses, and the engagement of compulsive actions or rituals to alleviate associated anxiety and fear [1]. Obsessions in OCD are defined by the intrusion of thoughts, images, or sounds, which are unwelcome and unavoidable, whereas compulsions are the rule-bound behaviours that one performs to obstruct unwanted obsessions [2]. Additionally, cognitive inflexibility is commonly presented in OCD patients [3]. Patients with OCD experience a significant reduction in their quality of life (QoL) [4] across different cultures, for example Asian [5] and North American [6] cultures.

According to the 5th edition of The Diagnostic and Statistical Manual of Mental Disorder (DSM-V), the diagnosis of OCD is met when there are all four of the following features: the presence of compulsions, obsessions, or both; severe enough to be time-consuming/cause significant distress/significant impairment; the disturbance is not due to physiological effects of a substance or another medical condition; the disturbance is not better explained by the symptoms of other mental disorders [1]. Obsessions and compulsions are defined consistently across the 10th edition of the International Statistical Classification of Diseases (ICD-10) [7] and DSM-V. ICD-11, which has already been released by the World Health Organization (WHO) and will soon replace ICD-10, has many revisions and additions for OCD. First, OCD has been reclassified from “Neurotic, Stressed Related, and Somatoform Disorders” to “Obsessive Compulsive and Related Disorders (OCRD)”, which is similar to DSM-V [1]. Second, the requirement of a 2-week duration of symptoms has been abolished because of the absence of robust data [8]. In parallel with DSM-V, ICD-11 specifiers now include the level of insight [1].

The lifetime prevalence of OCD is estimated to be up to 2% [9]; OCD is ranked sixth in the Global Burden of Disease (GBD) and is associated with a 10-fold increased risk of death from suicide compared to the general population [10,11,12]. Many studies have shown that during adulthood, OCD occurs more frequently in women than men [13,14], although others found the disorder’s prevalence to be higher in boys during childhood [15]. 

Despite both obsessions and compulsions being distinctive attributes of OCD, some patients mainly present with obsessive thoughts but limited or no compulsive actions [16]. Even though there is some evidence of obsessions appearing as sensory impressions, mostly in visual imagery, research on auditory imagery, such as musical obsession, remains scarce [17]. Nonetheless, this clinical feature of OCD is listed under “miscellaneous obsessions” in the Yale–Brown Obsessive Compulsive Symptom Checklist (Y-BOCS) [18] and defined as the intrusion of musical fragments, sounds, or tunes, inciting anxiety that is tedious to restrain [19]. Roughly one hundred occurrences of musical obsessions have been reported worldwide, predominantly in young adults [19].

### 1.2. Music-Related Symptoms in OCD Patients 

Many individuals also experience involuntary musical imagery, or “earworms”, which occur when a repetitive piece of music is spontaneously and involuntarily recalled inside the “mind’s ear” [20]. Despite the possibility of disturbances, unlike obsessions, involuntary musical imagery is not ego-dystonic, where one’s behaviours conflict with one’s beliefs [20]. This phenomenon is often observed in the non-psychiatric population [21]. An online survey found that 89.2% of survey responders experienced it weekly, and 33.2% daily [22]. Furthermore, although Sacks [23] and Levitin [24] suggested that involuntary musical imagery is closely correlated to OCD considering its overlapping characteristics with musical obsession, Negishi and Sekiguchi [21] believe that involuntary musical imagery is not necessarily a symptom of OCD. 

Music hallucinations, or musical hallucinations, also count as involuntary musical imagery [25,26]. Auditory hallucinations are defined by the hearing of songs, music, or sounds in an absence of an external musical source [23,27]. However, the affected individual is not aware that the music hallucinations are unreal and only exist within their mind. Musical hallucinations were once regarded as a rare phenomenon. However, recent studies have shown their commonness among the psychiatric population, especially in OCD patients, with a prevalence of up to 41% [28]. The number rises to 50% in individuals with other comorbid mental disorders such as schizophrenia or bipolar disorder [22]. 

Concerning music hallucinations, the term ‘musical pseudohallucination’ has often been used despite its controversial definition. Despite the similarity in name, pseudohallucinations are recognised by the affected person as being subjective and unreal. Pseudohallucinations seem to share more standard features with musical obsessions than musical hallucinations [29]. Both phenomena occur in the personal space (inner ears) and independently of one’s will. Although obsessions are also experienced from the personal space, Lewis has described three essential features for diagnosing obsessions: subjective compulsion, resistance to them, and the preservation of insight (26). Table 1 summarises the characteristics of hallucinations, pseudohallucinations, and musical obsession.

OCD symptoms have a significant impact on patients and the life of their families. Compared to some other psychiatric illnesses (e.g., panic disorder and social phobias), individuals with OCD have lower mean QoL scores [31]. Additionally, suicidal thoughts and behaviours are frequent in people with OCD [32]. Apart from the devastating impact of OCD on one’s life, more than half of their families/carers report performing rituals together with the patient [33], leading to the development of depression and low QoL in family members and carers [34].

### 1.3. Treatments for OCD and Music Therapy

The current treatments for OCD may not always result in a cure, but can be helpful in bringing symptoms under control, alleviating its burden on everyday life and delaying the aggravation of the condition. According to the National Institute for Health and Care Excellence (NICE) guidelines [35], specialist services should directly engage and provide treatments for OCD. Additionally, the information of the disorder, its course, and treatments should be well communicated with the patient’s families and carers (34). Psychological and pharmacological approaches are the two main types of treatment for OCD. Medication with selective serotonin reuptake inhibitors (SSRIs) has been preferred to treat OCD, with sertraline most commonly used [36]. Evidence-based psychotherapy, including cognitive behavioural therapy (CBT) with exposure and response prevention and cognitive therapy elements, have been reported to be effective for many individuals with OCD [37]. However, although SSRIs’ success rate is up to 60% [38], CBT’s efficacy is routinely inconsistent between people. Even the combination of both treatment approaches only reaches up to 70% effectiveness [38]. Thus, additional therapeutic approaches are needed.

Generally, music therapy has shown positive impacts on people with mental health problems. As a catalyst of change, music therapy can reduce the symptoms of various disorders and foster overall well-being [38]. Some studies, for example, Atiwannapat et al. [39], have found that individual or group music therapy, either active or receptive, as an adjunct to standard medication and psychological treatment, have some effect in reducing depressive scores in major depressive disorder (MDD) [40]. Furthermore, a systematic review by Testa et al. summarised studies that reported improvements in food consumption and symptom reduction in anorexia nervosa patients with the help of listening to classical music and singing [41]. Considering the comorbidity of OCD and other mental disorders (e.g., general anxiety disorders and eating disorders), the therapeutic advantages of music may also be used to alleviate symptoms of OCD or anxious and depressive symptoms in people with OCD [40,41,42,43,44]. Several empirical studies have suggested the benefits of music therapy on OCD. For example, receptive music therapy helped reduce obsessive symptoms with comorbid anxiety and depression [42]. A replication of this research reported similar findings where receptive music therapy was beneficial in easing the seriousness of both obsessive and compulsive symptoms [43]. Furthermore, improvisational music therapy might also alleviate symptoms in OCD patients [44].

### 1.4. Aim of This Systematic Review

With this review, we sought to explore the connections between music and OCD systematically.

## 2. Materials and Methods

### 2.1. Search Strategy

This systematic review was conducted according to the Preferred Reporting Items for Systematic Reviews and Meta-Analyses (PRISMA) guidelines [44]. The study protocol was published using the Open Science Framework (OSF): https://osf.io/a2rqm/ [accessed on 17 October 2021].

Studies were identified through Boolean operators by combining free-text words. This method is considered one of the best ways to yield more accurate search results [45]. Following various scoping searches, PubMed, Web of Science, and PsycInfo were chosen to extract relevant studies from inception until October 2021. The key search terms for PubMed were: (“obsessive compulsive disorder” OR “obsessive compulsive disorder” OR “OCD”) AND (“music” OR “music therapy”). The Web of Science search terms were: TS = (music OR music therapy) AND TS = (obsessive compulsive disorder OR obsessive compulsive disorder OR OCD), and for PsycInfo via Ovid, search terms were: (“obsessive compulsive disorder” OR “obsessive compulsive disorder” OR “OCD” AND (“music” OR “music therapy”). A.T. also conducted supplementary hand searching and citation-chaining by appraising the reference lists of selected articles.

### 2.2. Inclusion and Exclusion Criteria

Inclusion criteria:The studies are published in English or have an English-language abstract available;Participants of studies who were diagnosed with or had symptoms of OCD or had obsessive compulsive (OC) personality traits and experienced music hallucinations or musical obsessions or received music therapy; studies must be OCD- and music-related;Original studies or case reports.

Exclusion criteria:No results or no clinical outcomes regarding the use of music, the effects of music or the effects of any therapy on musical symptoms are reported. This is, for example, the case for articles where music therapy is only mentioned as a treatment for a specific patient or as part of the therapeutic program in a hospital without reporting its effects;Systematic reviews or meta-analyses;The study involved dance or art therapy primarily, but not specifically the effect of music;The published article was not an original study or case report;Studies which mainly explored the mental health of musicians, music students and music therapists in general, without investigating OCD or OCD symptoms;If music was not part of the symptoms or music therapy was not applied;Studies where data were not obtained from the participants directly, for example, case studies of historic composers;The studies used animal testing;The studies described mainly a hospital or a therapy program;The studies reported OCD patients as the minority unless results for the sub-group were provided;Participants were diagnosed with OCD but (1) did not have musical obsessions, (2) did not have musical hallucinations, or (3) did not receive music therapy as a treatment;The presentation of musical obsessions and musical hallucinations occurred mainly in mental disorders other than OCD;The full text of the article was not available.

### 2.3. Study Selection and Data Extraction

The results from both searches were imported into Endnote20—a reference management software. All duplicate papers were removed by first using the “Find Duplicates” function in Endnote20 and then manually by the first author, T.P.A.T. Eligible studies were screened according to their titles and abstracts by T.P.A.T. The second screening was performed by the last author (H.H.) following an independent PubMed, Web of Science, and PsycINFO search. Candidate papers were reviewed in full text and then extracted utilising Microsoft Excel (version 2021) by T.P.A.T., and then verified by H.H. for accuracy. Both authors had agreed on the relevant information for extraction prior to starting the process. The main extracted details included citation details, study characteristics, types of treatment, and main outcomes.

### 2.4. Data Analysis

After the extraction of study details, the articles were thematically arranged based on the study design and types of intervention by both T.P.A.T. and H.H. The findings were then reported accordingly.

## 3. Results

### 3.1. Included Studies

Following two searches, 145 candidate papers were identified, and 3 studies were added through hand-searching and reference-chaining. After the removal of duplicates, a total of 97 abstracts were screened and 90 available full-text articles were assessed for inclusion. A total of 27 studies met the full eligibility criteria and were chosen for analysis. These studies contained data of *n* = 650 patients or study participants. Figure 1 shows a PRISMA diagram describing the results of the search strategy and reasons for exclusion. Table 2 summarises all the publications that met the eligibility criteria and were included in this systematic review.

### 3.2. Treatment of Musical Obsessions and Hallucinations in People Diagnosed with OCD

A total of 19 of the 27 identified papers were case reports describing the presentation of musical obsessions or hallucinations in OCD patients and their treatment. Patients reported in 11 case reports had their symptoms managed by pharmacological interventions, whereas 8 of them were treated with a combination of both pharmacological and non-pharmacological approaches.

#### 3.2.1. Pharmacological Interventions

Eleven out of twenty-seven papers were case reports of OCD patients with musical obsessions or hallucinations who received pharmacological treatment with SSRIs, antipsychotics, benzodiazepines, or tricyclic antidepressants (TCA). Six of these case reports described treatments with SSRIs which led to a reduction in OCD symptoms, including musical obsessions and music hallucinations [29,48,49,50,51,52].

Bergman et al. [29] reported a case series of six elderly individuals whose total Y-BOCS scores were significantly reduced after treatment with escitalopram. The choice of using SSRIs was made after unsuccessful treatment with antipsychotic agents, and more notably because of the emergence of antipsychotic-induced akathisia and Parkinsonism [29]. Similarly, Focseneau [46] highlighted the efficacy of escitalopram on the management of OCD symptomatology. Interestingly, a slight improvement in music hallucinations was observed when adding the atypical antipsychotic drug risperidone into the care plan. However, when increasing the dose of risperidone, extrapyramidal symptoms appeared.

A 32-year-old female patient in Orjuela-Rojas and Rodriguez’s paper [52] experienced a reduction in her music obsessions after 12 weeks of using fluvoxamine. Compared to pre-treatment, her Y-BOCS scores decreased from 33 to 13, indicating a 60% reduction from severe to mild OCD [52]. Similarly, Mendhekar and Andrade [51] presented a younger patient (age 22) who also self-reported an improvement after treatment with fluvoxamine.

A report by Güçlü [48] found that a combination of SSRI (fluvoxamine) and the antipsychotic quetiapine effectively reduced OC symptoms and the intensity of musical obsessions. A female patient (44 years old), self-admitted to the psychiatric department due to the appearance of instrumental tunes that she had never previously heard [48], was first treated with both fluvoxamine and quetiapine, with a particularly high dose of fluvoxamine at 200 mg/d. After two months, despite the OC symptoms having faded, the musical hallucinations remained present. When quetiapine was increased to 200 mg/d, the intensity of music hallucinations reduced [48]. Her 1-year outcome was favourable, with no more symptoms of OCD or music hallucinations [48].

Islam, Scarone, and Gambini [49] presented the case of a 51-year-old woman who received paroxetine to manage her intrusive musical obsessions. Her OCD onset was at the age of 15, and she had been prescribed many types of SSRIs, including sertraline, paroxetine, and clomipramine, which helped improve her OCD symptoms [49]. She also underwent CBT but stopped after 1.5 years because she thought the therapy was not beneficial [49]. Three months after the diagnosis of hearing loss, she experienced musical obsession for the first time. After 20 weeks on paroxetine, her Y-BOCS scores had reduced from 36 to 16 [49].

A case series produced by Maheran [50] showed conflicting results regarding the efficacy of SSRIs. One out of three cases (a 56-year-old woman) reported the disappearance of music hallucinations post-treatment with citalopram, whereas the other two cases (61-year-old widow and 78-year-old man) were resistant to fluvoxamine and citalopram, respectively [50]. However, the 61-year-old patient was able to manage her music hallucinations by masking the symptoms with other music, whereas the symptoms surprisingly disappeared after colon-cancer surgery in the 78-year-old man [50].

Two reports described treatment with the TCA clomipramine for OCD patients with musical obsessions [53,56]. Three patients from these two studies were under the age of 30 [53,56]. Matsui et al. [16] reported a decrease in Y-BOCS scores and depressive symptoms after 12 weeks of treatment for case 1 (20-year-old male) and eight weeks for case 2 (28-year-old male). Surprisingly, in Pfizer and Andrade’s report, the symptoms of musical obsession remained unchanged after treatment with clomipramine and alprazolam [53].

Gomibuchi et al. [47] presented two young male students (18 and 19 years old) who experienced musical obsessions. After being treated with diazepam, the five-year outcome of the 19-year-old student was favourable and there were no recurrent symptoms [47]. However, the second case reported the reappearance of musical obsessions 5 years post-treatment with bromazepam [47].

Zungu-Dirwayi et al. [54] reported treatment-refractory cases of both SSRIs and antipsychotics. Their first case was a 59-year-old woman who experienced intrusive and unwanted tunes that interfered with her daily life [53]. Due to her treatment resistance with SSRIs, comprising fluoxetine, paroxetine, citalopram, and clomipramine, she was prescribed risperidone. Unfortunately, treatment with risperidone was not successful. The second patient refused a psychopharmacological intervention.

#### 3.2.2. Combinations of Pharmacological and Non-Pharmacological Interventions

Eight of nineteen papers reported the combination of pharmacological and non-pharmacological treatment in managing musical obsession and music hallucinations in OCD patients.

Aneja, Nebhinani, and Grover [55] reported a case of a 20-year-old male diagnosed with OCD and musical obsession. He was first prescribed desvenlafaxine. Despite a slight decrease in OCD symptoms after eight weeks of treatment, he was readmitted to the hospital due to a lack of improvement [55]. He was then given the SSRI escitalopram with additional thought-stopping behavioural therapy. Finally, a total remission of symptoms was recorded after 12 weeks [55]. Additionally, the two reports by Naskar, Victor, Nath and Choudhury [58], and Prahajah et al. [30] presented two male patients (25 and 21 years old, respectively) who also received a combination treatment of SSRIs and thought-stopping techniques. The 25-year-old-patient’s Y-BOCS scores decreased from 26 to 9 after discharge [58], whereas the 21-year-old patient experienced a decrease in OC symptoms after using SSRIs together with antipsychotics and recorded reduced musical obsession frequency after practicing the thought-stopping technique for 6 weeks [30].

A case report conducted by Chauhan, Shah and Grover [17] presented a 35-year-old female patient with OCD and obsessive auditory imagery symptoms. Her pharmacological treatment included various SSRIs: sertraline, fluoxetine, fluvoxamine and clomipramine, either alone or in combination [17]. Initially, the symptoms were slightly reduced but then reoccurred, further worsening the disorder [17]. However, after introducing various behavioural therapies in addition to clomipramine, her Y-BOCS scores had improved by 75% [17].

Liikkanen and Raaska [22] reported a case of musical obsessions comorbid with schizophrenia. The patient received different types of medication—risperidone, fluvoxamine, escitalopram, clozapine—as part of her treatment for paranoid schizophrenia, anxiety, and depression, which resulted in drug-induced obsessive involuntary musical imagery [22]. The musical obsessions worsened with clozapine treatment, but because clozapine treatment was necessary to control her psychotic symptoms, she received CBT to treat her musical obsessions [22]. According to the authors, this cognitive approach helped her gain more insight into her condition, which eventually helped to improve her mental state, as measured with the PANSS scale [22].

Two published independent reports of OCD patients with musical obsession and musical hallucinations showed symptom improvements after receiving a combined treatment of SSRIs and a non-pharmacological intervention [56,59]. Matta, Ribas and Carod-Artal [56] described three patients (two men and one woman) with musical obsession symptoms. Case 3 only received the pharmacological treatment compared to the other two who had combined treatments of an SSRI plus CBT. All three patients reported a reduction in symptoms [56]. Similarly, the 30-year-old reported in Saha’s paper [59] was prescribed fluvoxamine combined with individual psychotherapy, exposure prevention therapy, and systematic desensitisation [59]. After one month of treatment, his Y-BOCS scores decreased significantly and nearly reached complete remission at the 6-month follow-up [59].

Lastly, Nath et al. [57] presented a case of a 22-year-old male patient diagnosed with OCD, tic disorder, and musical obsessions. He was first treated with clomipramine, which helped alleviate the OC symptoms, but not his musical obsessions [57]. It was only when he was introduced to various behavioural techniques—thought-stopping, exposure and response prevention, and Morita therapy—that he self-reported a slight improvement in musical obsessions and a significant reduction in other OC features [57].

### 3.3. Music Perception and Musical Obsessions

#### 3.3.1. Music Perception and Musical Obsessions in People with Obsessive-Compulsive Personality Traits

Buse et al. [60] investigated whether harmonic expectancy violations caused incitement of “not-just-right experiences” and if the emotional appraisal of those violations and the duration necessary to notice them was related to “general experiences of incompleteness”. The response time (RT) to the disharmonic chord was lower than to the harmonic sequences, and the experience of incompleteness correlated with the presence of OC symptoms [60].

Negishi and Sekiguchi [21] investigated the impact of individual characteristics on the recurrence and emotional traits of involuntary musical imagery. Using the experience sampling method (ESM), participants were asked to record their emotional characteristics at the moment of involuntary musical imagery’s occurrence [21]. In addition, personality traits and music expertise were measured. Although the OC tendencies were reported to positively influence the involuntary musical imagery frequency, they negatively impacted the pleasantness of involuntary musical imagery’s experience and the extent to which they liked the music [21]. Moreover, despite the description of experiencing involuntary musical imagery as pleasant, the level of pleasantness varied between individuals [21].

#### 3.3.2. Music Perception and Musical Obsessions in People with OCD

An fMRI-study conducted by Buse and Roessner [61] found that patients with OCD exhibited more brain activation than healthy controls in harmonic conditions and lower activation in disharmonic conditions. Therefore, these findings might support the relationship between OCD and an attention-misallocation to auditory stimulus that would ordinarily be processed unconsciously [61].

Nielzen and Cesarec [62,63] showed that people diagnosed with OCD are more emotionally responsive to music than healthy subjects [62] and experience more tension when listening to music compared to patients with schizophrenic, affective and anxiety disorders. [63]. In both groups, participants were exposed to seven pieces of orchestral music and their emotional experiences were measured using the three-factor semantic differential scales. OCD patients rated all music more intensely compared to healthy subjects, as well as patients with other psychiatric illnesses [63]. Despite the finding of enhanced sensitivity in OCD patients, their representation in the studies was relatively small, with only 12 patients with OCD among 107 participants. Additionally, other comorbid disorders presenting with OCD might influence the rating results, which was not considered in both studies [62,63].

### 3.4. The Influence of Music and Music Therapy on Patients with OCD

Two studies explored the influence of therapeutic listening to music as an adjunct to standard treatment for patients with OCD [42,43].

Abdulah, Alhakem and Piro [43] investigated the effect of listening to music in 36 adult patients with OCD who were randomly allocated to the music therapy group or a control group. The participants allocated to music therapy were instructed to listen daily to one of seven prepared music pieces for a three-month period at any time during the day or night for 50 min along with their standard pharmacological treatment. The selected music tracks were relaxing music, sleep music and meditation music by Peder B. Helland, a music composer from Norway, produced with a wide variety of instruments, including a piano, guitar, flute, harp, violin and cello. The control group only received standard treatment [43]. After three months, patients in the music therapy group had significantly less severe obsessive and compulsive symptoms [43].

Bidabadi and Mehryar [42] examined the effect of receptive music therapy, which consisted of listening to selected tracks of Iranian classical music followed by discussions with experienced psychiatrists. A total of 30 patients with OCD were included into the study and allocated to music therapy plus treatment as usual or standard treatment only. Over a span of four weeks, each individual in the music therapy group took part in 12 sessions of music therapy. They found that music therapy resulted in a greater decrease in the total obsessive scores than standard treatment. Significant between-group differences were identified for checking and slowness, but not for washing or responsibility. Compared to treatment as usual, music therapy also exhibited a greater reduction in depressive and anxiety symptoms [42].

Ciambella et al. [64] investigated the effect of music therapy as adjunctive therapy on resocialising compared to standard treatment in 24 patients with various psychiatric disorders, which were mood disorders, obsessive-compulsive disorder or schizophrenia. Participants were allocated to two groups: Group 1 received improvisational music therapy only; the 13 patients in Group 2 altered between improvisational music therapy and music listening sessions [64]. However, there was only one patient with OCD in each treatment group. The combination of improvisational and receptive music therapy sessions yielded a higher resocialising effect than receptive music therapy alone when both groups were compared [64].

## 4. Discussion

### 4.1. Overview of the Main Results of This Systematic Review and Their Significance

This review aimed to explore the connection between OCD or obsessive-compulsive symptoms and music. The systematic literature review yielded 27 eligible articles including 650 study participants on three different subtopics: intrusive musical perceptions in people with OCD and their treatment, the peculiarities of how music is perceived in people with obsessive-compulsive symptoms and personality traits or OCD diagnosis, and the effects of music therapy with people diagnosed with OCD.

Regarding repetitive and intrusive musical perceptions in people with OCD, we found eleven papers reporting the sole use of pharmacological treatment [16,29,46,47,48,49,50,51,52,53,54]. Most of the reported patients with musical obsessions responded to SSRIs, for example, escitalopram [17,29,46,58] fluvoxamine [17,22,30,48,50,51,52,58,59] and the TCA clomipramine [16,17,53,57,58]. However, there were some papers that reported non-responses to SSRI treatment. For instance, in the report by Mahendran [50], patients with MHs did not respond to either fluvoxamine or citalopram. Additionally, musical OCD symptoms may persist even if other OCD symptoms improve [54]. Thus, in summary, the literature on the treatment of musical hallucinations and musical imagery comprises several case reports on SSRI treatment. However, the results are inconsistent and randomised controlled studies (RCTs) are lacking.

Further case presentations have reported on the combination of different classes of drugs or a psychopharmacological drug with the addition of a specific psychotherapy. For example, the combination of SSRIs and antipsychotics resulted in improvement of musical OCD symptoms in some cases [46,48]; however, there were cases presented with adverse or no effects [29,54]. In Nath et al. [57], SSRIs were only able to alleviate OCD symptoms but not musical obsessions. Contrastingly, Liikkanen and Raaska [22] reported the appearance of drug-induced obsessive involuntary musical imagery after being treated with a wide range of medications such as fluvoxamine, escitalopram, risperidone and clozapine. Liikkanen and Raaska [22] reported the use of additional psychotherapeutic techniques in OCD patients with comorbid psychosis. This approach led to a reduction in auditory imagery symptoms. A combination of pharmacotherapy and psychotherapy may thus be beneficial for OCD patients with musical obsessions or music hallucinations. These findings were also aligning with the results of previous studies [19].

Even though it is unusual for systematic reviews to incorporate clinical case reports, this review decided to include them because of the clinical relevance of these symptoms in people with OCD. Thus, clinicians and specifically music therapists should be aware that music might be or even become part of the disorder. Additionally, the list of various medications used in people with OCD and musical obsessions or hallucinations provides medical doctors with ideas what medications they could prescribe for similar patients.

This review has also identified the relationship between music and obsessive-compulsive traits as well as the clinical diagnosis of OCD. In the identified studies, people with OC traits tended to be more emotionally responsive to music, sounds or tunes [60]. For instance, people with higher OC traits are more likely to experience involuntary musical imagery [21]. Similarly, people with OCD are more sensitive to music than healthy subjects or patients diagnosed with other disorders. This might be explained by the relationship between OCD and an attention-misallocation to an auditory stimulus. Thus, it appears natural to draw a link between OCD and music-based interventions and therapy.

Despite the limited number of studies on music therapy in treating OCD, the identified clinical trials [42,43,64] found some evidence for the effectiveness of music therapy in reducing OCD symptom frequency and severity. The majority of studies used receptive music therapy, whereas only one paper compared receptive music therapy with a combined listening and improvisational music therapy [64]. Both clinical studies on receptive music therapy used listening to classical music and found significant beneficial effects on OCD symptoms [42,43]. In one of the studies, patients listened to preselected classical music on their own [43], in the second music therapy sessions consisted of listening to music and a subsequent conversation with a psychiatrist [42]. Ciambella et al. [64] found that the combination of receptive and improvisational music therapy was more beneficial to the patients with various psychiatric disorders. However, in this study, there was only one single patient with OCD in each treatment group. Overall, the limited number of studies and the lack of a rigorous RCTs do not allow for drawing firm conclusions. However, receptive music therapy with classical music should be considered for future research.

### 4.2. Limitations

This systematic review has several limitations. Firstly, despite the comprehensiveness of the literature search, unpublished studies, studies not published in English or studies published in journals not included in PubMed, Web of Science or PsycInfo might have been missed, which could have led to a lack of relevant studies. Additionally, most of the selected articles were mainly conducted in countries with a high socio-economic status, which reduces its generalisability to locations with fewer resources.

Secondly, due to the heterogeneity of the studies, meta-analyses and uniform assessments of the studies’ qualities were not possible. The design of the obtained studies included case reports, cross sectional, and longitudinal studies. Most of the published papers related to musical OCD symptoms were case reports; therefore, the evidence for the described therapeutic approaches is very limited and drawing firm conclusions is not possible.

Thirdly, one of the crucial methodological problems of this review is the use of the terms. Most of the articles included did not specify how they used the terms “music hallucinations”, “musical obsession”, “musical pseudohallucinations”, and “involuntary musical imagery.” Thus, the same clinical symptom might have been called “music hallucination” in one article and “involuntary musical imagery” in another article. Therefore, future research demands clear definitions and operationalisation of research terms and outcomes.

Lastly, regarding the use of music therapy, only three studies had a longitudinal study design [42,43,64]. Of these three studies, the study by Ciambella et al. [64] included only one patient with OCD in each treatment group. Group sample sizes in the remaining two RCTs were small, with *n* = 15 [42] and *n* = 17 [43] people in the music therapy groups. Thus, despite the RCT study design, these studies should be viewed as pilot studies and conclusions derived from these two studies should be drawn with caution. Due to the scarcity of longitudinal music therapy studies and the limited evidence for the utilisation of music therapy in OCD, we were unable to propose general recommendations for music therapy in people with OCD based on this systematic review.

### 4.3. Future Directions

Future studies on the therapeutic use of music in helping patients with OCD should take the peculiarities of how music is received by people with OCD into account. Our systematic review has not yielded studies on qualitative aspects of the patients’ perspective of and attitudes towards music, similarly to how they have been performed in other areas such as eating disorders [65]. To obtain specific answers from future research, the population should be well defined in terms of age (children, adolescents, adults or elderly people), severity and symptomatology. Research exploring the format and type of music therapy (individual or group music therapy, receptive or active music therapy) would help in identifying its benefits on people with OCD. The description of music-based interventions in any scientific article should follow the reporting guidelines for music-based interventions and include information about the intervention, its content duration and frequency, information about who selected the music, the music utilised, the music delivery method and the interventionist [66]. Additionally, the group of study participants, the control group and the outcomes should be well defined in a future study protocol. This approach will help to compare different interventions. Furthermore, assessing OCD symptoms comprehensively and longitudinally should include music-related symptoms. Longitudinal studies with an RCT design and appropriate group sizes in each treatment arm will be necessary to draw firm conclusions and to develop data-based recommendations on the use of music to help people with OCD.

## 5. Conclusions

We have identified three different connections between music and OCD: First, the perception of music can appear as a symptom of OCD. Published case reports suggest that these musical obsessions may be treated with SSRIs or a combination of an SSRI with psychotherapy or with an atypical antipsychotic. However, there is no evidence from clinical studies to further substantiate this therapeutic approach; Second, studies regarding the peculiarities of music perception in people with obsessive-compulsive personality traits or OCD have shown an increased desire for harmony and alignment in these people when listening to music; Third, music may have therapeutic potential for people with OCD. However, the level evidence for the use of music listening and music therapy in the treatment of OCD is low, and the specific approaches (active vs. passive) and the settings (group vs. individual therapy; inpatient vs. outpatient) remain unclear.

## Figures and Tables

**Figure 1 ijerph-18-11799-f001:**
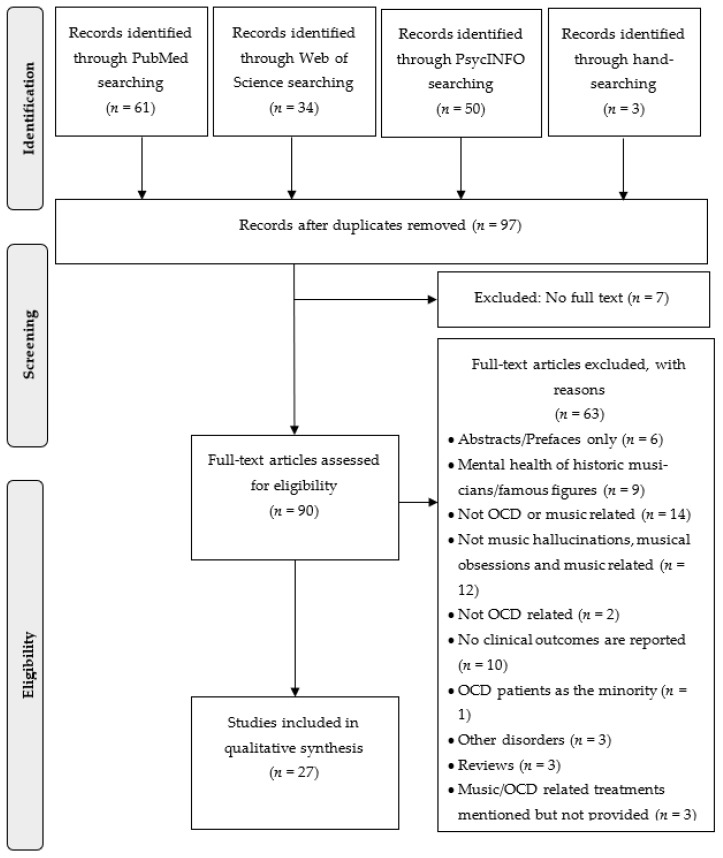
Flow-chart of literature search according to PRISMA.

**Table 1 ijerph-18-11799-t001:** Characteristics of musical hallucinations, pseudohallucinations and obsessions [30].

Musical Hallucinations	Musical Pseudohallucinations	Musical Obsessions
Corporeal, complete, sensory-rich; Projected in external space; Independent of the subject’s will; No insight that the auditory sensation is subjective and unreal; No compulsion; No resistance.	Imaginary; Projected in internal space; Independent of the subject’s will; Have insight that the auditory sensation is subjective and unreal; No compulsion; No resistance.	Figurative, subjective space; Heard in the past; No voluntary control; Associated compulsion possible; Insight; Resistance.

**Table 2 ijerph-18-11799-t002:** Summary of publications included in the systematic review which met the eligibility criteria.

No.	Author	Sample and Group Size (*n*)	Total *N*	Study Design	Questionnaires and Research Methods	Types of Treatment	Main Outcomes	Statistical Significance of Main Results
(1) Case reports on the musical symptoms in patients with OCD and their treatment								
(1a) Pharmacological treatments								
1	Bergman et al. (2014) [29]	Elderly patients diagnosed with OCD and music hallucinations; female (*n* = 4) and male (*n* = 2)	6	Case series	Mini-mental status examination Y-BOCS adapted to music hallucinations	Escitalopram (10 to 20 mg/d), perphenazine (4 to 8 mg/d), risperidone (1 to 2 mg/d)	Decrease in the severity and the duration of musical hallucinations, decrease in Y-BOCS total score under 10 to 20 mg/d escitalopram.	∆Y-BOCS total score before and after escitalopram *p* < 0.01 (mean Y-BOCS before: 13.2 ± 0.9; after: 7.8 ± 2.8)
2	Focseneanu (2015) [46]	Elderly female patient, 71 years old, diagnosed with OCD and music hallucinations (*n* = 1)	1	Case report	HAMD, HAM-A, MMSE, Y-BOCS	Escitalopram (10 mg/d), lorazepam (10 mg/d), risperidone—starting 1 mg/day for 2 weeks, then 2 mg/d for another 2 weeks, and finally 3 mg/d	Slight improvement in symptoms at 2 mg/d risperidone; a daily dose of 3 mg/d risperidone induced extrapyramidal symptoms and cognitive impairment.	N/A
3	Gomibuchi (2000) [47]	Male patients diagnosed with hearing music as an obsession (*n* = 2)	2	Case series	N/A	C1: Diazepam (6 mg/d) C2: Bromazepam (6 mg/d)	Diazepam did not lead to symptom improvement; symptoms diminished after 1 week 6 mg/d bromazepam.	N/A
4	Güçlü (2013) [48]	44-year-old female patient diagnosed with OCD and music hallucinations (*n* = 1)	1	Case report	N/A	Fluvoxamine (up to 200 mg/d within 2 months) quetiapine (200 mg/d)	Reduction in OCD or musical hallucinations during treatment with fluvoxamine and quetiapine.	N/A
5	Islam et al. (2014) [49]	51-year-old female patient diagnosed with OCD, otosclerosis with musical obsessions (*n* = 1)	1	Case report	N/A	Paroxetine (20 mg/d to 40 mg/d)	YBOCS scores decreased from 26 to 15; HAM-D scored from 40 to 6 and HAM-A scores from 14 to 4 after 8 weeks of treatment at 3 months follow-up visit.	N/A
6	Mahendran (2007) [50]	Case 1: 61-year-old widow diagnosed with depression with music hallucination symptoms (*n* = 1) Case 2: 78-year-old man who experienced music hallucinations (*n* = 1) Case 3: 56-year-old woman diagnosed with OCD with major depression and music hallucination symptoms (*n* = 1)	3	Case series	N/A	C1: Fluvoxamine (up to 150 mg/d) C2: Citalopram (no dose was given) C3: Citalopram (20 to 40 mg/d)	C1: Musical hallucinations continued but the patient masked the symptoms with singing and/or playing other music. C2: Symptoms disappeared 10 months after a major surgery for colon cancer. C3: OCD and depressive symptoms disappeared under 40 mg/d citalopram.	N/A
7	Matsui et al. (2003) [16]	Case 1: 20-year-old male diagnosed with OCD and dysthymic disorder with musical symptoms (*n* = 1) Case 2: 28-year-old male diagnosed with OCD with musical symptoms (*n* = 1)	2	Case series	Y-BOCS SDS	C1: Haloperidol (1 to 12 mg/d) Risperidone (6 mg/d) Clomipramine (25 to 150 mg/d) C2: Haloperidol (6 mg/d) Clomipramine (25 to 150 mg/d)	Both ratings of Y-BOCS and depressive symptoms decreased within 12 weeks (case 1) and 8 weeks (case 2), both at 150 mg clomipramine per day	N/A
8	Mendhekar and Andrade (2009) [51]	22-year-old male patient diagnosed with OCD (*n* = 1)	1	Case report	Y-BCOS	Fluvoxamine (200 mg/d), clomipramine (75 mg/d)	Subjective improvement.	N/A
9	Orjuela-Rojas and Rodriguez (2018) [52]	32-year-old female patient diagnosed with OCD, musical obsessions (*n* = 1)	1	Case report	Y-BOCS The Beck Depression Inventory HAM-A	Fluvoxamine (200 mg/d), paroxetine (50 mg/d), fluoxetine (60 to 80 mg/d)	Fluoxetine (60 mg/d) reduced obsessive symptoms by 60% and improved quality of life (Y-BOCS scores). Headache at 80 mg of fluoxetine per day.	N/A
10	Pfizer and Andrade (1999) [53]	25-year-old female patient diagnosed with OCD and musical obsession (*n* = 1)	1	Case report	N/A	Clomipramine (25 mg three times daily), alprazolam (0.25 mg three times daily), antipyretic drugs	Musical obsessions remained unchanged.	N/A
11	Zungu-Dirway et al. (1999) [54]	Case 1: 59-year-old woman presented with musical obsessions (*n* = 1) Case 2: 29-year-old woman presented with musical obsessions (*n* = 1)	2	Case series	N/A	C1: Fluoxetine, paroxetine, citalopram, clomipramine, risperidone (no dose reported) C2: Refuse any types of intervention (offered pharmacological)	C1: Treatment resistance. C2: No improvement in symptoms.	N/A
(1b) Combinations of pharmacological and non-pharmacological interventions								
1	Aneja et al. (2015) [55]	20-year-old male patient diagnosed with OCD and musical obsession (*n* = 1)	1	Case report	N/A	Desvenlafaxine (50 mg/d) Escitalopram (10 to 30 mg/d), behaviour therapy: Thought-stopping	A slight improvement over a period of 8 weeks under desvenlafaxine (50 mg/d). With the increase in escitalopram to 30 mg/d and supervised thought-stopping, the patient achieved remission after 12 weeks.	N/A
2	Chauhan et al. (2010) [17]	35-year-old female patient diagnosed with OCD and obsessive auditory imagery	1	Case report	Y-BOCS	Sertraline (up to 200 mg/d) fluoxetine (80 mg/d) fluvoxamine (300 mg/d) clomipramine (300 mg/d); behaviour therapy: exposure, response prevention, vivo exposure	Y-BOCS score improved under 300 mg clomipramine per day by 75%; SSRIs had no significant effect.	N/A
3	Liikkanen and Raaska (2013) [22]	36-year-old female patient diagnosed with schizophrenia with obsessive musical hallucinations (*n* = 1)	1	Case report	N/A	CBT-inspired instrumental music adjunct to pharmacological treatment:olanzapine (5 to 10mg/d), risperidone (up to 4mg/d), fluoxetine (20mg/d), fluvoxamine (50 to 100mg/d, escitalopram (10mg/d), clozapine (5 to 75 mg/d)	Frequency of and disturbance due to symptoms decreased under clozapine only at 75 mg/d.	N/A
4	Matta et al. (2012) [56]	Case 1: 57-year-old woman diagnosed with OCD and experiences of musical obsessions (*n* = 1) Case 2: 25-year-old man with history of OCD and current experiences of musical obsessions (*n* = 1) Case 3: 47-year-old man diagnosed with OCD and experiences of musical obsessions (*n* = 1)	3	Case series	N/A	C1 and 2: Sertraline (100 to 150 mg/d) + CBT C3: Citalopram (20 to 40 mg/d), Alprazolam (0.5 mg per 12 h)	3 cases: Symptoms improved moderately.	N/A
5	Nath et al. (2013) [57]	22-year-old male patient diagnosed with OCD, comorbid tic disorder, and musical obsessions (*n* = 1)	1	Case report	N/A	Olanzapine (10 mg/d) clomipramine (50 to 150 mg/d), flupentixol (1 mg/d) + thought-stopping, response-prevention, Morita therapy	Improvement in musical obsessions and reduction in other obsessive and compulsive features at 150 mg/d clomipramine + 1 mg/d flupentixol and thought-stopping, response-prevention, Morita therapy.	N/A
6	Naskar et al. (2017) [58]	25-year-old male patient diagnosed with OCD and musical obsession (*n* = 1)	1	Case report	HAM-A and HAM-D Y-BOCS	Escitalopram (20 to 40 mg/d). No doses were reported for desvenlafaxine, clomipramine, fluvoxamine, risperidone Thought-stopping	Reduction in intensity and frequency of symptoms with 40 mg escitalopram per day and behaviour therapy.	N/A
7	Praharaj et al. 2009 [30]	21-year-old male patient diagnosed with OCD with persistent musical obsessions (*n* = 1)	1	Case report	N/A	Fluvoxamine (up to 300 mg/d), risperidone (2 mg/d) Thought-stopping technique	Reduction in both the duration and frequency of musical obsessions.	N/A
8	Saha (2012) [59]	30-year-old male patient diagnosed with OCD and musical obsession.	1	Case report	Y-BOCS	Fluvoxamine (200 mg/d), fluoxetine (60 mg/d), lithium (900 mg/d) Individual psychotherapy, exposure prevention therapy, and systematic desensitisation	Subjective improvement after 4 weeks and scored lower on Y-BOCS after 8 weeks.	N/A
(2) Studies on the perception of music in people with obsessive compulsive personality traits or an OCD diagnosis								
(2a) Studies in people with obsessive compulsive personality traits								
1	Buse et al. (2015) [60]	Healthy young adults; Females (*n* = 46); Males (*n* = 19)	64	RCT, Harmonic expectancy violation paradigm: harmonic condition vs. disharmonic condition	OCI-R, OCTCDQ-R, OCTCDQ-GR, Harm avoidance and general experiences of incompleteness	N/A	The response time (RT) to disharmonic chord sequences was significantly shorter than the RT to harmonic chord sequences.	∆RT to harmonic and RT to disharmonic was significantly correlated to incompleteness score (*p* = 0.05)
2	Negishi and Sekiguchi (2020) [21]	University students aged 18 to 24: Females (*n* = 58), Males (*n* = 43)	101	Experience sampling study	Obsessive compulsive tendencies (Japanese adaptation of Maudsley Obsessional Compulsive Inventory, the Padua Inventory, the indecisiveness scale) A short form of Big Five (Japanese version/adaptation) Gold-MSI PACO	N/A	Obsessive compulsive tendencies were positively correlated with the involuntary musical imagery, negatively correlated with pleasantness of involuntary musical imagery experience and the extent to likability of the music heard internally. Obsessive compulsive personality traits and music expertise were significantly associated with involuntary musical imagery occurrence.	Effect of compulsive washing and pleasantness (*p* < 0.01), liking the music (*p* < 0.05). Effect of openness and liking the music (*p* < 0.01). Effect of intrusive thoughts and involuntary musical imagery occurrences (*p* < 0.01). Effect of singing abilities and involuntary musical imagery occurrences (*p* < 0.01).
(2b) Studies in people with an OCD diagnosis								
1	Buse and Roessner (2016) [61]	Boys with OCD (*n* = 21) versus healthy controls (*n* = 29)	50	Harmonic expectancy violation paradigm: harmonic condition vs. disharmonic condition	fMRI, CY-BOCS, OCI-R, OCTCDQ-R, ZWIK	N/A	Boys with OCD exhibited increased activation compared to healthy controls in the harmonic condition and decreased condition in the disharmonic condition.	Response time (RT) to disharmonic chord sequences was significantly faster than the RT to harmonic chord sequences (*p* < 0.001). Error rates in disharmonic condition were significantly smaller than in the harmonic condition (*p* < 0.001)
2	Nielzén and Cesarec (1982a) [62]	Inpatients with schizophrenic psychosis (*n* = 22); depressive psychosis (*n* = 12); manic psychosis (*n* = 10); obsessive neurosis (*n* = 12); depressive neurosis (*n* = 20); anxiety neurosis (*n* = 12); hysterical neurosis (*n* = 19) versus normal subject (*n* = 100)	207	As stimuli, 7 short pieces of orchestral music were exposed to psychiatric patients and normal subjects.	Assessment of emotional experience: tension-relaxation, gaiety-gloom, attraction-repulsion	N/A	Obsessive compulsive group rated all music as more tense, more sensitive to tense music than normal group.	N/A
3	Nielzen and Cesarec (1982b) [63]	Inpatients with schizophrenic psychosis (*n* = 22); depressive psychosis (*n* = 12); manic psychosis (*n* = 10); obsessive neurosis (*n* = 12); depressive neurosis (*n* = 20); anxiety neurosis (*n* = 12); hysterical neurosis (*n* = 19)	107	As stimuli, 7 short pieces of orchestral music were exposed to psychiatric patients and normal subjects.	Assessment of emotional experience: tension-relaxation, gaiety-gloom, attraction-repulsion	N/A	Obsessive compulsive group rated all music as more sensitive to tense music compared to schizophrenia, depressive, and anxiety groups.	N/A
(3) Studies on the effect of music and music therapy for patients with OCD								
1	Abdulah et al. (2019) [43]	Females with OCD (*n* = 19); Males with OCD (*n* = 17)	36	RCT; Music tracks (relaxing, sleep and meditation music) composed by Peder B. Helland vs. regular treatment; parallel group.	YBOCS	Music therapy; frequency: 7 50 min tracks/per day; duration: 3 months	Obsessive compulsive symptoms and obsessions and compulsions were lower in group treated with music therapy compared to regular treatment alone.	N/A
2	Ciambella, et at. (2019) [64]	Patients of a psychiatric day hospital. Group 1 received add-on therapy on 14 improvisational music therapy (*n* = 11). Group 2 received 7 improvisational music therapy sessions alternated with 7 listening music therapy therapy session (*n* = 13). In each group was one patient with OCD.	24	Add-on therapy improvisational music therapy session vs. improvisational music therapy session alternated with listening music therapy session.	Assess the efficacy of music therapy through nine variables: method of approaching the instruments, method of handling the instruments, vocal production method, interaction frequency, sharing instruments, gaze interaction, facial expression, tension, and movement.	Music therapy: Frequency: Group 1: 14 improvisational music therapy sessions/per week Group 2: 7 improvisational + 7 listening music session/per week Duration: 6 months	In both groups, good response from OCD patients.	N/A
3	Shiranibidabadi and Mehryar (2015) [42]	Outpatients with OCD who live in Iran (*n* = 30)	30	RCT; standard treatment + music therapy vs. standard treatment only; music therapy: receptive or individual or group	BAI, BDI-SF, MOCI	Music therapy Frequency and Duration: 12 sessions over 1 month	Music therapy results in greater decrease in obsessive score, anxiety, and depressive symptoms.	∆ anxiety, distress: *p* < 0.001 ∆ obsessive score: *p* < 0.001

Abbreviations: BAI: Beck Anxiety Inventory; BDI-SF: Beck Depression Inventory—Short Form; CY-BOCS: Children’s Yale–Brown Obsessive-Compulsive Scale; fMRI: Functional Magnetic Resonance Imagine; Gold-MSI: Goldsmith Musical Sophistication Index; HAM-A: Hamilton Anxiety Scale; HMA-D: Hamilton Depression Raring Scale; mg/d: milligram per day; MMSE: mini-mental state examination; MOCI: Maudsley Obsessive-Compulsive Inventory; OCI-R: Obsessive-Compulsive Inventory; OCTCDQ-GR: Obsessive-Compulsive Trait Core Dimensions Questionnaire—Revision in German; OCTCDQ-R: Obsessive-Compulsive Trait Core Dimensions Questionnaire—Revision; PACO: Personal Analytics Companion; RCT: randomised control trial; SDS: Zung’s Self-Rating Depression Scale; Y-BOCS: Yale–Brown Obsessive-Compulsive Scale; ZWIK: Zwangsinventar für Kinder und Jugendliche/ a German questionnaire to dimensionally assess paediatric OC symptoms was completed by all participants and their parents. ∆: Difference.

## Data Availability

Data availability is not applicable to this article because no new data were created or analysed in this study.

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
