# Peer review of "A Systematic Review of Scientific Studies and Case Reports on Music and Obsessive-Compulsive Disorder"

_ijerph, 2021, doi:10.3390/ijerph182211799_

Round 1
Reviewer 1 Report
This review is interesting and important for understanding effects of music on obsessive-compulsive disorder. I suggest some revisions in order to improve the manuscript.
Method section
- Authors mentioned that this review conducted according to the PRISMA guideline (line 146). Thus, protocol of this systematic review should be presented in "Method section".
e.g. https://doi.org/10.3390/jcm10143087
- I recommend authors to supplement detail inclusion and exclusion of intervention (e.g. types of music, program, practitioner, and definition of music therapy) in Method section.
Result section
- What is "No treatments were" in figure 1? (line 217)
- total number is not matching in figure 1: 74-46=28 (line 217)
- Regarding patient, intervention, control group and outcome (PICO), explanation of intervention (music therapy) is important in this review. Information on the types of music, program, duration, frequency, and practitioner(qulification) of music therapy. (Table 3) In particular, the informations on music therapy in references [21, 42, 60-64] are limited.
Discussion
- Also, I recommend authors to propose a standardization for music therapy basd on a systematic review.
Author Response
Response to reviewer 1:
This review is interesting and important for understanding the effects of music on obsessive-compulsive disorder. I suggest some revisions in order to improve the manuscript.
Method section
- The authors mentioned that this review was conducted according to the PRISMA guideline (line 146). Thus, the protocol of this systematic review should be presented in the "Method section". e.g. https://doi.org/10.3390/jcm10143087
We thank the reviewer for this comment; and we have added the link to the protocol into paragraph 2.1. of the methods section: “This systematic review was conducted according to the Preferred Reporting Items for Systematic Reviews and Meta-analyses (PRISMA) guidelines [44]. The study protocol was published using the Open Science Framework (OSF): https://osf.io/a2rqm/.”
- I recommend authors supplement detailed inclusion and exclusion of intervention (e.g. types of music, program, practitioner, and definition of music therapy) in the Method section.
Before starting the systematic review, we assumed that the number of published studies on music therapy in OCD would be limited. Therefore, we wanted to include all available studies and did not specify inclusion and exclusion criteria further. Indeed, the literature search yielded only three studies on the effect of music and music therapy for patients with OCD (Abdulah et al. 2019; Ciambella, et at. 2019; Shiranibidabadi & Mehryar 2015). We believe that the number of clinical trials should not be reduced further by focusing on specific music therapies or programs. Result section
- What is "No treatments were" in figure 1? (line 217)
The correct sentence is: “No treatments were provided (n = 3).” We have amended the figure accordingly.
- We have added the information in Figure 1. total number is not matching in figure 1: 74-46=28 (line 217)
We apologize for the miscalculation. We have amended the figure.
- Regarding patient intervention, control group and outcome (PICO), an explanation of intervention (music therapy) are important in this review. Information on the types of music, program, duration, frequency, and practitioner(qualification) of music therapy. (Table 3) In particular, the information on music therapy in references [21, 42, 60-64] are limited.
We have included the available information on the type of music, its frequency and duration in Table 3. Studies [62, 62] used music as a stimulus to explore the variation of emotions in music from diagnostic groups. No other information regarding duration and frequency was reported in the included articles. However, we took the reviewer’s concern into account and wrote in the discussion: “The description of the music-based intervention in any scientific article should follow the reporting guidelines for music-based interventions and include information about intervention, its content duration and frequency, information about who selected the music, music utilized, music delivery method, and the interventionist [68]. Additionally, the group of study participants, the control group and the outcomes should be well defined in a future study protocol. This approach will help to compare different interventions.”
Discussion
- Also, I recommend authors propose a standardization for music therapy based on a systematic review.
Due to the scarcity of longitudinal music therapy studies and therefore limited evidence for the utilization of music therapy in OCD, we were unable to propose a standardization for music therapy based on our systematic review. Thus, we have added to our discussion: “Due to the scarcity of longitudinal music therapy studies and the limited evidence for the utilization of music therapy in OCD, we were unable to propose general recommendations for music therapy in people with OCD based on this systematic review.”

Reviewer 2 Report
The authors conducted a systematic review of music-related variables and OCD. They found that the literature is broadly centred on three main areas, namely musical obsessions, musical perception, and the use of music therapy in treatment. This is a very interesting topic. However, I have many major concerns about the strength of the paper.
Overall manuscript:
- The manuscript would benefit from some proofreading. For example, in the first paragraph of the Introduction, it should read “unwelcome” and not “unwelcomed”. In the same paragraph, “e.g.” which appears outside of parentheses should be spelt out (“for example”). On p. 3, “an OCD’s symptom” should instead read “an OCD symptom” or “a symptom of OCD”. The authors should thoroughly proofread their paper to check for more of such errors.
- Too many acronyms are used, which makes it difficult for the reader to follow. Other than commonly-known acronyms (OCD, SSRIs, CBT, RT, etc.), acronyms (e.g., MH, INMI, MT, NJRE, GEOI, BOLD) should be avoided.
Introduction:
- The second to fourth paragraphs provide details about OCD. While informative, this is not directly relevant to the study of OCD and music, which is the focus of this paper. These paragraphs should either be greatly summarised, or otherwise linked clearly to music.
- The overall flow/structure of the Introduction can be improved. Currently, the authors talk about OCD, then talk about musical imagery and other related things, then move back to OCD, then move to music therapy. This is disruptive and difficult to follow.
Methods:
- Given that both OCD and music-related variables are widely studied in psychology, it is surprising that the authors did not include PsycInfo as a database from which to retrieve literature. Please justify this decision, or otherwise re-conduct the search with the addition of PsycInfo.
- The authors must justify why there was a lower limit on the year of publication (1996 for PubMed and 1997 for Web of Science) during their search. The authors also must justify why this limit is not consistent across the two databases.
- More details should be provided about the hand-searching and citation-chaining processes. Who conducted these and how?
- In the inclusion criteria, “The report of results or clinical outcomes.” is extremely vague. Please rephrase/elaborate. What results or clinical outcomes are considered relevant or appropriate?
- Please justify this exclusion criterion: “The studies were explored predominantly on the mental health of musicians, music students and music therapists”. It is possible for these samples to also have OCD, so it is unclear why these samples are automatically excluded.
- Please justify this exclusion criterion: “The studies described mainly a hospital or a therapy program”. I was under the impression that music therapy studies would be eligible?
- Please specify how duplicate records were identified and removed (p. 5).
- The data extraction section (p. 5) is completely lacking. All coded variables should be clearly defined and described. Details should also be provided about who conducted the data extraction (was it also AT and HH? This should be clearer). If multiple people conducted the data extraction, an agreement rate and range should be reported. If only one person conducted the data extraction, this should be addressed as a limitation.
- I would appreciate more details about how the thematic arrangement of included records was conducted, and by whom (p. 5).
Results:
- The PRISMA flowchart is cut off and has missing arrows.
- It is unclear how the bulk of the findings on pharmacological interventions (pp. 5–6) are relevant to the study of OCD and music.
Discussion:
- The Discussion is quite long and unfocused. The authors should consider summarising and integrating their points better.
Author Response
Response to reviewer 2:
Overall manuscript:
The authors conducted a systematic review of music-related variables and OCD. They found that the literature is broadly centred on three main areas, namely musical obsessions, musical perception, and the use of music therapy in treatment. This is a very interesting topic. However, I have many major concerns about the strength of the paper.
- The manuscript would benefit from some proofreading. For example, in the first paragraph of the Introduction, it should read “unwelcome” and not “unwelcomed”. In the same paragraph, “e.g.” which appears outside of parentheses should be spelt out (“for example”). On p. 3, “an OCD’s symptom” should instead read “an OCD symptom” or “a symptom of OCD”. The authors should thoroughly proofread their paper to check for more of such errors.
Thank you very much for your feedback. One of the authors, Briana Applewhite, a native English speaker, has proofread the manuscript carefully.
- Too many acronyms are used, which makes it difficult for the reader to follow. Other than commonly-known acronyms (OCD, SSRIs, CBT, RT, etc.), acronyms (e.g., MH, INMI, MT, NJRE, GEOI, BOLD) should be avoided.
We have converted the less commonly known acronyms into the full elaboration of words.
Introduction:
- The second to fourth paragraphs provide details about OCD. While informative, this is not directly relevant to the study of OCD and music, which is the focus of this paper. These paragraphs should either be greatly summarised or otherwise linked clearly to music.
- The overall flow/structure of the Introduction can be improved. Currently, the authors talk about OCD, then talk about musical imagery and other related things, then move back to OCD, then move to music therapy. This is disruptive and difficult to follow.
We agree with these two pieces of feedback on our introduction. Thus, we have restructured and reduced the information related to OCD.
Methods:
- Given that both OCD and music-related variables are widely studied in psychology, it is surprising that the authors did not include PsycInfo as a database from which to retrieve literature. Please justify this decision, or otherwise re-conduct the search with the addition of PsycInfo.
We agree with the feedback given by the reviewer. Thus, we have re-conducted the search with the addition of the PsycInfo database and included it in our review.
- The authors must justify why there was a lower limit on the year of publication (1996 for PubMed and 1997 for Web of Science) during their search. The authors also must justify why this limit is not consistent across the two databases.
We apologize for any confusion. We have mistakenly added the dates where the search should be conducted from inception until now. This has been updated accordingly.
- More details should be provided about the hand-searching and citation-chaining processes. Who conducted these and how?
We have added more details about the process as follows: “AT also conducted supplementary hand-searching and citation-chaining by going through the reference lists of selected articles.”
- In the inclusion criteria, “The report of results or clinical outcomes.” is extremely vague. Please rephrase/elaborate. What results or clinical outcomes are considered relevant or appropriate?
We have deleted this inclusion criterium and added to the exclusion criteria “No results or no clinical outcomes regarding the use of music, the effects of music or the effects of any therapy on musical symptoms are not reported. This is, for example, the case for articles where music therapy is only mentioned as a treatment for a specific patient or as part of the therapeutic program in a hospital without reporting its effects.”
- Please justify this exclusion criterion: “The studies were explored predominantly on the mental health of musicians, music students and music therapists”. It is possible for these samples to also have OCD, so it is unclear why these samples are automatically excluded.
We apologize for the lack of clarity. We have rephrased it as follows: “Studies which explored mainly the mental health of musicians, music students and music therapists in general, if music was not part of the symptoms or music therapy was not applied. Studies where data were not obtained from the participants directly, for example, case studies of historic composers”
- Please justify this exclusion criterion: “The studies described mainly a hospital or a therapy program”. I was under the impression that music therapy studies would be eligible?
In order to clarify this, we have combined this criterium with the one about results and outcomes. As explained above, the new criterium reads: “No results or no clinical outcomes regarding the use of music, the effects of music or the effects of any therapy on musical symptoms are not reported. This is, for example, the case for articles where music therapy is only mentioned as a treatment for a specific patient or as part of the therapeutic program in a hospital without reporting its effects.”
- Please specify how duplicate records were identified and removed (p. 5).
We used the “Find Duplicates” function in Endnote20 to identify and remove duplicate records. A manual screening also took place after removing duplicate articles from Endnote20. We have explained this in our revised methods section.
- The data extraction section (p. 5) is completely lacking. All coded variables should be clearly defined and described. Details should also be provided about who conducted the data extraction (was it also AT and HH? This should be clearer). If multiple people conducted the data extraction, an agreement rate and range should be reported. If only one person conducted the data extraction, this should be addressed as a limitation.
We agree with the reviewer’s feedback. Thus, we have changed the section accordingly.
“The results from both searches were imported into Endnote20 — a reference management software. All duplicate papers were removed by first using the “Find Duplicates” function in Endnote20 and then manually by AT. Eligible studies were screened according to their titles and abstracts by the first author (AT). The second screening was done by the second author (HH) following independently searching PubMed, Web of Science, and PsycINFO. Candidate papers were reviewed in full text and then extracted utilizing Microsoft Excel (version 2021) by AT and then be verified by HH for accuracy. Both authors had agreed on the relevant items for extraction prior to starting the process. The main extracted details included citation details, study characteristics, types of treatment, and main outcomes.”
- I would appreciate more details about how the thematic arrangement of included records was conducted, and by whom (p. 5).
We have changed the section as follows: “​​After extraction of study details, the articles were thematically arranged based on the study design and types of intervention by both AT and HH. The findings then were reported accordingly.”
Results:
- The PRISMA flowchart is cut off and has missing arrows.
We have amended the PRISMA flowchart and added the missing details.
- It is unclear how the bulk of the findings on pharmacological interventions (pp. 5–6) are relevant to the study of OCD and music.
We included those articles because of the clinical relevance of these symptoms in people with OCD. We have therefore written in the revised version of our discussion: “Even though it is unusual for systematic reviews to incorporate clinical case reports, this review decided to include them because of the clinical relevance of these symptoms in people with OCD. Thus, clinicians and specifically music therapists should be aware that music might be or even become part of the disorder. Additionally, the list of various medications used in people with OCD and musical obsessions or hallucinations pro-vides medical doctors with ideas what medications they could prescribe for similar patients.”
Discussion:
- The Discussion is quite long and unfocused. The authors should consider summarising and integrating their points better.
We agree with the reviewer, and we have therefore shortened and structured the discussion. However, we also added a few sentences to respond to comments and questions raised by the reviewers.

Round 2
Reviewer 2 Report
The authors have sufficiently address my comments
Author Response
We thank the reviewer for the helpful comments which we have addressed in our first revision of the manuscript. We are grateful for the reviewer's positive feedback in response to our revised version that we have sufficiently addressed these comments.
Additionally, we thank the reviewer for encouraging us to revise the English language of our article further in order to improve the description of the applied methods, the presentation of the results and our conclusions.
We have used "track changes" throughout the manuscript to indicate our amendments. As the reviewer did not have additional specific requests, and as the reviewer has already stated that we have sufficiently addressed the specific suggestions made in review round 1, we hope that our manuscript is now suitable for publication.